# DARTS: Differentiable Architecture Search

**Hanxiao Liu**[*]
CMU
hanxiaol@cs.cmu.com

**Karen Simonyan**
DeepMind
simonyan@google.com

**Yiming Yang**
CMU
yiming@cs.cmu.edu

## ABSTRACT

This paper addresses the scalability challenge of architecture search by formulating the task in a differentiable manner. Unlike conventional approaches of applying evolution or reinforcement learning over a discrete and non-differentiable search space, our method is based on the continuous relaxation of the architecture representation, allowing efficient search of the architecture using gradient descent. Extensive experiments on CIFAR-10, ImageNet, Penn Treebank and WikiText-2 show that our algorithm excels in discovering high-performance convolutional architectures for image classification and recurrent architectures for language modeling, while being orders of magnitude faster than state-of-the-art non-differentiable techniques. Our implementation has been made publicly available to facilitate further research on efficient architecture search algorithms.

## 1 INTRODUCTION

Discovering state-of-the-art neural network architectures requires substantial effort of human experts. Recently, there has been a growing interest in developing algorithmic solutions to automate the manual process of architecture design. The automatically searched architectures have achieved highly competitive performance in tasks such as image classification (Zoph & Le, 2017; Zoph et al., 2018; Liu et al., 2018b;a; Real et al., 2018) and object detection (Zoph et al., 2018).

The best existing architecture search algorithms are computationally demanding despite their remarkable performance. For example, obtaining a state-of-the-art architecture for CIFAR-10 and ImageNet required 2000 GPU days of reinforcement learning (RL) (Zoph et al., 2018) or 3150 GPU days of evolution (Real et al., 2018). Several approaches for speeding up have been proposed, such as imposing a particular structure of the search space (Liu et al., 2018b;a), weights or performance prediction for each individual architecture (Brock et al., 2018; Baker et al., 2018) and weight sharing/inheritance across multiple architectures (Elsken et al., 2017; Pham et al., 2018b; Cai et al., 2018; Bender et al., 2018), but the fundamental challenge of scalability remains. An inherent cause of inefficiency for the dominant approaches, e.g. based on RL, evolution, MCTS (Negrinho & Gordon, 2017), SMBO (Liu et al., 2018a) or Bayesian optimization (Kandasamy et al., 2018), is the fact that architecture search is treated as a black-box optimization problem over a discrete domain, which leads to a large number of architecture evaluations required.

In this work, we approach the problem from a different angle, and propose a method for efficient architecture search called DARTS (Differentiable ARchiTecture Search). Instead of searching over a discrete set of candidate architectures, we relax the search space to be continuous, so that the architecture can be optimized with respect to its validation set performance by gradient descent. The data efficiency of gradient-based optimization, as opposed to inefficient black-box search, allows DARTS to achieve competitive performance with the state of the art using orders of magnitude less computation resources. It also outperforms another recent efficient architecture search method, ENAS (Pham et al., 2018b). Notably, DARTS is simpler than many existing approaches as it does not involve controllers (Zoph & Le, 2017; Baker et al., 2017; Zoph et al., 2018; Pham et al., 2018b; Zhong et al., 2018), hypernetworks (Brock et al., 2018) or performance predictors (Liu et al., 2018a), yet it is generic enough handle both convolutional and recurrent architectures.

The idea of searching architectures within a continuous domain is not new (Saxena & Verbeek, 2016; Ahmed & Torresani, 2017; Veniat & Denoyer, 2017; Shin et al., 2018), but there are several major

---

[*]Current affiliation: Google Brain.

distinctions. While prior works seek to fine-tune a specific aspect of an architecture, such as filter shapes or branching patterns in a convolutional network, DARTS is able to learn high-performance architecture building blocks with complex graph topologies within a rich search space. Moreover, DARTS is not restricted to any specific architecture family, and is applicable to both convolutional and recurrent networks.

In our experiments (Sect. 3) we show that DARTS is able to design a convolutional cell that achieves $2.76 \pm 0.09\%$ test error on CIFAR-10 for image classification using 3.3M parameters, which is competitive with the state-of-the-art result by regularized evolution (Real et al., 2018) obtained using three orders of magnitude more computation resources. The same convolutional cell also achieves 26.7% top-1 error when transferred to ImageNet (mobile setting), which is comparable to the best RL method (Zoph et al., 2018). On the language modeling task, DARTS efficiently discovers a recurrent cell that achieves 55.7 test perplexity on Penn Treebank (PTB), outperforming both extensively tuned LSTM (Melis et al., 2018) and all the existing automatically searched cells based on NAS (Zoph & Le, 2017) and ENAS (Pham et al., 2018b).

Our contributions can be summarized as follows:

- We introduce a novel algorithm for differentiable network architecture search based on bilevel optimization, which is applicable to both convolutional and recurrent architectures.
- Through extensive experiments on image classification and language modeling tasks we show that gradient-based architecture search achieves highly competitive results on CIFAR-10 and outperforms the state of the art on PTB. This is a very interesting result, considering that so far the best architecture search methods used non-differentiable search techniques, e.g. based on RL (Zoph et al., 2018) or evolution (Real et al., 2018; Liu et al., 2018b).
- We achieve remarkable efficiency improvement (reducing the cost of architecture discovery to a few GPU days), which we attribute to the use of gradient-based optimization as opposed to non-differentiable search techniques.
- We show that the architectures learned by DARTS on CIFAR-10 and PTB are transferable to ImageNet and WikiText-2, respectively.

The implementation of DARTS is available at https://github.com/quark0/darts

## 2 DIFFERENTIABLE ARCHITECTURE SEARCH

We describe our search space in general form in Sect. 2.1, where the computation procedure for an architecture (or a cell in it) is represented as a directed acyclic graph. We then introduce a simple continuous relaxation scheme for our search space which leads to a differentiable learning objective for the joint optimization of the architecture and its weights (Sect. 2.2). Finally, we propose an approximation technique to make the algorithm computationally feasible and efficient (Sect. 2.3).

### 2.1 SEARCH SPACE

Following Zoph et al. (2018); Real et al. (2018); Liu et al. (2018a;b), we search for a computation cell as the building block of the final architecture. The learned cell could either be stacked to form a convolutional network or recursively connected to form a recurrent network.

A cell is a directed acyclic graph consisting of an ordered sequence of $N$ nodes. Each node $x^{(i)}$ is a latent representation (e.g. a feature map in convolutional networks) and each directed edge $(i, j)$ is associated with some operation $o^{(i,j)}$ that transforms $x^{(i)}$. We assume the cell to have two input nodes and a single output node. For convolutional cells, the input nodes are defined as the cell outputs in the previous two layers (Zoph et al., 2018). For recurrent cells, these are defined as the input at the current step and the state carried from the previous step. The output of the cell is obtained by applying a reduction operation (e.g. concatenation) to all the intermediate nodes.

Each intermediate node is computed based on all of its predecessors:

$$x^{(j)} = \sum_{i<j} o^{(i,j)}(x^{(i)}) \qquad (1)$$

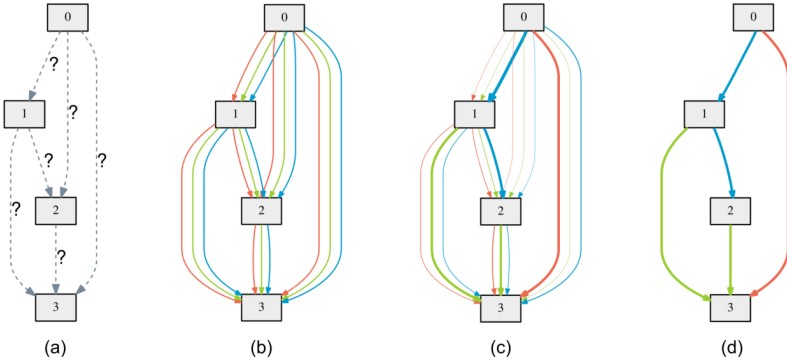

Figure 1: An overview of DARTS: (a) Operations on the edges are initially unknown. (b) Continuous relaxation of the search space by placing a mixture of candidate operations on each edge. (c) Joint optimization of the mixing probabilities and the network weights by solving a bilevel optimization problem. (d) Inducing the final architecture from the learned mixing probabilities.

A special *zero* operation is also included to indicate a lack of connection between two nodes. The task of learning the cell therefore reduces to learning the operations on its edges.

## 2.2 CONTINUOUS RELAXATION AND OPTIMIZATION

Let $\mathcal{O}$ be a set of candidate operations (e.g., convolution, max pooling, *zero*) where each operation represents some function $o(\cdot)$ to be applied to $x^{(i)}$. To make the search space continuous, we relax the categorical choice of a particular operation to a softmax over all possible operations:

$$\bar{o}^{(i,j)}(x) = \sum_{o \in \mathcal{O}} \frac{\exp(\alpha_o^{(i,j)})}{\sum_{o' \in \mathcal{O}} \exp(\alpha_{o'}^{(i,j)})} o(x) \tag{2}$$

where the operation mixing weights for a pair of nodes $(i, j)$ are parameterized by a vector $\alpha^{(i,j)}$ of dimension $|\mathcal{O}|$. The task of architecture search then reduces to learning a set of continuous variables $\alpha = \{\alpha^{(i,j)}\}$, as illustrated in Fig. 1. At the end of search, a discrete architecture can be obtained by replacing each mixed operation $\bar{o}^{(i,j)}$ with the most likely operation, i.e., $o^{(i,j)} = \text{argmax}_{o \in \mathcal{O}} \; \alpha_o^{(i,j)}$. In the following, we refer to $\alpha$ as the (encoding of the) architecture.

After relaxation, our goal is to jointly learn the architecture $\alpha$ and the weights $w$ within all the mixed operations (e.g. weights of the convolution filters). Analogous to architecture search using RL (Zoph & Le, 2017; Zoph et al., 2018; Pham et al., 2018b) or evolution (Liu et al., 2018b; Real et al., 2018) where the validation set performance is treated as the reward or fitness, DARTS aims to optimize the validation loss, but using gradient descent.

Denote by $\mathcal{L}_{train}$ and $\mathcal{L}_{val}$ the training and the validation loss, respectively. Both losses are determined not only by the architecture $\alpha$, but also the weights $w$ in the network. The goal for architecture search is to find $\alpha^*$ that minimizes the validation loss $\mathcal{L}_{val}(w^*, \alpha^*)$, where the weights $w^*$ associated with the architecture are obtained by minimizing the training loss $w^* = \text{argmin}_w \mathcal{L}_{train}(w, \alpha^*)$.

This implies a bilevel optimization problem (Anandalingam & Friesz, 1992; Colson et al., 2007) with $\alpha$ as the upper-level variable and $w$ as the lower-level variable:

$$\min_{\alpha} \quad \mathcal{L}_{val}(w^*(\alpha), \alpha) \tag{3}$$

$$\text{s.t.} \quad w^*(\alpha) = \text{argmin}_w \; \mathcal{L}_{train}(w, \alpha) \tag{4}$$

The nested formulation also arises in gradient-based hyperparameter optimization (Maclaurin et al., 2015; Pedregosa, 2016; Franceschi et al., 2018), which is related in a sense that the architecture $\alpha$ could be viewed as a special type of hyperparameter, although its dimension is substantially higher than scalar-valued hyperparameters such as the learning rate, and it is harder to optimize.

---

**Algorithm 1:** DARTS – Differentiable Architecture Search

---

Create a mixed operation $\bar{o}^{(i,j)}$ parametrized by $\alpha^{(i,j)}$ for each edge $(i, j)$

**while** *not converged* **do**

    1. Update architecture $\alpha$ by descending $\nabla_\alpha \mathcal{L}_{val}(w - \xi \nabla_w \mathcal{L}_{train}(w, \alpha), \alpha)$

      ($\xi = 0$ if using first-order approximation)

    2. Update weights $w$ by descending $\nabla_w \mathcal{L}_{train}(w, \alpha)$

Derive the final architecture based on the learned $\alpha$.

---

## 2.3 Approximate Architecture Gradient

Evaluating the architecture gradient exactly can be prohibitive due to the expensive inner optimization. We therefore propose a simple approximation scheme as follows:

$$\nabla_\alpha \mathcal{L}_{val}(w^*(\alpha), \alpha) \tag{5}$$
$$\approx \nabla_\alpha \mathcal{L}_{val}(w - \xi \nabla_w \mathcal{L}_{train}(w, \alpha), \alpha) \tag{6}$$

where $w$ denotes the current weights maintained by the algorithm, and $\xi$ is the learning rate for a step of inner optimization. The idea is to *approximate $w^*(\alpha)$ by adapting $w$ using only a single training step*, without solving the inner optimization (equation 4) completely by training until convergence. Related techniques have been used in meta-learning for model transfer (Finn et al., 2017), gradient-based hyperparameter tuning (Luketina et al., 2016) and unrolled generative adversarial networks (Metz et al., 2017). Note equation 6 will reduce to $\nabla_\alpha \mathcal{L}_{val}(w, \alpha)$ if $w$ is already a local optimum for the inner optimization and thus $\nabla_w \mathcal{L}_{train}(w, \alpha) = 0$.

The iterative procedure is outlined in Alg. 1. While we are not currently aware of the convergence guarantees for our optimization algorithm, in practice it is able to reach a fixed point with a suitable choice of $\xi$[1]. We also note that when momentum is enabled for weight optimisation, the one-step unrolled learning objective in equation 6 is modified accordingly and all of our analysis still applies.

Applying chain rule to the approximate architecture gradient (equation 6) yields

$$\nabla_\alpha \mathcal{L}_{val}(w', \alpha) - \xi \nabla^2_{\alpha,w} \mathcal{L}_{train}(w, \alpha) \nabla_{w'} \mathcal{L}_{val}(w', \alpha) \tag{7}$$

where $w' = w - \xi \nabla_w \mathcal{L}_{train}(w, \alpha)$ denotes the weights for a one-step forward model. The expression above contains an expensive matrix-vector product in its second term. Fortunately, the complexity can be substantially reduced using the finite difference approximation. Let $\epsilon$ be a small scalar[2] and $w^\pm = w \pm \epsilon \nabla_{w'} \mathcal{L}_{val}(w', \alpha)$. Then:

$$\nabla^2_{\alpha,w} \mathcal{L}_{train}(w, \alpha) \nabla_{w'} \mathcal{L}_{val}(w', \alpha) \approx \frac{\nabla_\alpha \mathcal{L}_{train}(w^+, \alpha) - \nabla_\alpha \mathcal{L}_{train}(w^-, \alpha)}{2\epsilon} \tag{8}$$

Evaluating the finite difference requires only two forward passes for the weights and two backward passes for $\alpha$, and the complexity is reduced from $O(|\alpha||w|)$ to $O(|\alpha| + |w|)$.

**First-order Approximation** When $\xi = 0$, the second-order derivative in equation 7 will disappear. In this case, the architecture gradient is given by $\nabla_\alpha \mathcal{L}_{val}(w, \alpha)$, corresponding to the simple heuristic of optimizing the validation loss by assuming the current $w$ is the same as $w^*(\alpha)$. This leads to some speed-up but empirically worse performance, according to our experimental results in Table 1 and Table 2. In the following, we refer to the case of $\xi = 0$ as the first-order approximation, and refer to the gradient formulation with $\xi > 0$ as the second-order approximation.

## 2.4 Deriving Discrete Architectures

To form each node in the discrete architecture, we retain the top-$k$ strongest operations (from distinct nodes) among all non-zero candidate operations collected from all the previous nodes. The strength of an operation is defined as $\frac{\exp(\alpha_o^{(i,j)})}{\sum_{o' \in \mathcal{O}} \exp(\alpha_{o'}^{(i,j)})}$. To make our derived architecture comparable with

---

[1]A simple working strategy is to set $\xi$ equal to the learning rate for $w$'s optimizer.

[2]We found $\epsilon = 0.01 / \|\nabla_{w'} \mathcal{L}_{val}(w', \alpha)\|_2$ to be sufficiently accurate in all of our experiments.

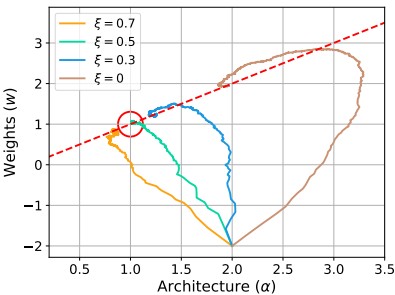

Figure 2: Learning dynamics of our iterative algorithm when $\mathcal{L}_{val}(w, \alpha) = \alpha w - 2\alpha + 1$ and $\mathcal{L}_{train}(w, \alpha) = w^2 - 2\alpha w + \alpha^2$, starting from $(\alpha^{(0)}, w^{(0)}) = (2, -2)$. The analytical solution for the corresponding bilevel optimization problem is $(\alpha^*, w^*) = (1, 1)$, which is highlighted in the red circle. The dashed red line indicates the feasible set where constraint equation 4 is satisfied exactly (namely, weights in $w$ are optimal for the given architecture $\alpha$). The example shows that a suitable choice of $\xi$ helps to converge to a better local optimum.

those in the existing works, we use $k = 2$ for convolutional cells (Zoph et al., 2018; Liu et al., 2018a; Real et al., 2018) and $k = 1$ for recurrent cells (Pham et al., 2018b).

The zero operations are excluded in the above for two reasons. First, we need exactly $k$ non-zero incoming edges per node for fair comparison with the existing models. Second, the strength of the zero operations is underdetermined, as increasing the logits of zero operations only affects the scale of the resulting node representations, and does not affect the final classification outcome due to the presence of batch normalization (Ioffe & Szegedy, 2015).

## 3 EXPERIMENTS AND RESULTS

Our experiments on CIFAR-10 and PTB consist of two stages, architecture search (Sect. 3.1) and architecture evaluation (Sect. 3.2). In the first stage, we search for the cell architectures using DARTS, and determine the best cells based on their validation performance. In the second stage, we use these cells to construct *larger* architectures, which we train from scratch and report their performance on the test set. We also investigate the transferability of the best cells learned on CIFAR-10 and PTB by evaluating them on ImageNet and WikiText-2 (WT2) respectively.

### 3.1 ARCHITECTURE SEARCH

#### 3.1.1 SEARCHING FOR CONVOLUTIONAL CELLS ON CIFAR-10

We include the following operations in $\mathcal{O}$: $3 \times 3$ and $5 \times 5$ separable convolutions, $3 \times 3$ and $5 \times 5$ dilated separable convolutions, $3 \times 3$ max pooling, $3 \times 3$ average pooling, identity, and *zero*. All operations are of stride one (if applicable) and the convolved feature maps are padded to preserve their spatial resolution. We use the ReLU-Conv-BN order for convolutional operations, and each separable convolution is always applied twice (Zoph et al., 2018; Real et al., 2018; Liu et al., 2018a).

Our convolutional cell consists of $N = 7$ nodes, among which the output node is defined as the depthwise concatenation of all the intermediate nodes (input nodes excluded). The rest of the setup follows Zoph et al. (2018); Liu et al. (2018a); Real et al. (2018), where a network is then formed by stacking multiple cells together. The first and second nodes of cell $k$ are set equal to the outputs of cell $k-2$ and cell $k-1$, respectively, and $1 \times 1$ convolutions are inserted as necessary. Cells located at the $1/3$ and $2/3$ of the total depth of the network are reduction cells, in which all the operations adjacent to the input nodes are of stride two. The architecture encoding therefore is $(\alpha_{normal}, \alpha_{reduce})$, where $\alpha_{normal}$ is shared by all the normal cells and $\alpha_{reduce}$ is shared by all the reduction cells.

Detailed experimental setup for this section can be found in Sect. A.1.1.

#### 3.1.2 SEARCHING FOR RECURRENT CELLS ON PENN TREEBANK

Our set of available operations includes linear transformations followed by one of tanh, relu, sigmoid activations, as well as the identity mapping and the *zero* operation. The choice of these candidate operations follows Zoph & Le (2017); Pham et al. (2018b).

Our recurrent cell consists of $N = 12$ nodes. The very first intermediate node is obtained by linearly transforming the two input nodes, adding up the results and then passing through a tanh activation

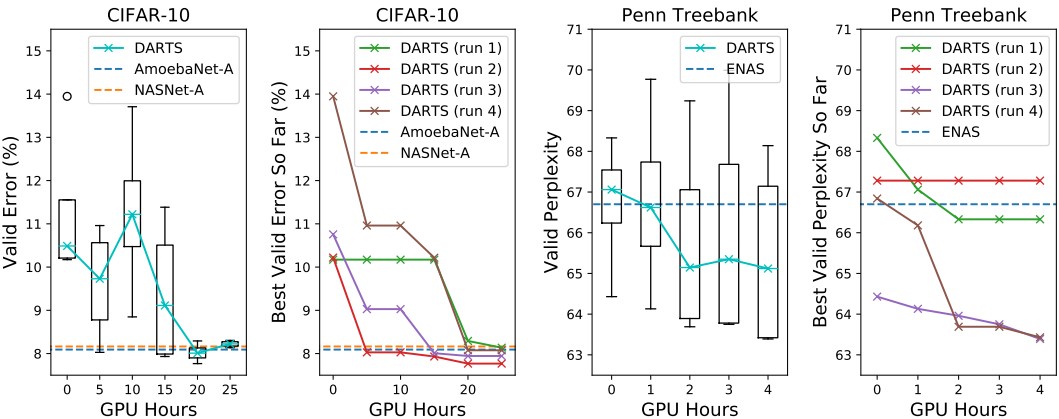

Figure 3: Search progress of DARTS for convolutional cells on CIFAR-10 and recurrent cells on Penn Treebank. We keep track of the most recent architectures over time. Each architecture snapshot is re-trained from scratch using the training set (for 100 epochs on CIFAR-10 and for 300 epochs on PTB) and then evaluated on the validation set. For each task, we repeat the experiments for 4 times with different random seeds, and report the median and the best (per run) validation performance of the architectures over time. As references, we also report the results (under the same evaluation setup; with comparable number of parameters) of the best existing cells discovered using RL or evolution, including NASNet-A (Zoph et al., 2018) (2000 GPU days), AmoebaNet-A (3150 GPU days) (Real et al., 2018) and ENAS (0.5 GPU day) (Pham et al., 2018b).

function, as done in the ENAS cell (Pham et al., 2018b). The rest of the cell is learned. Other settings are similar to ENAS, where each operation is enhanced with a highway bypass (Zilly et al., 2016) and the cell output is defined as the average of all the intermediate nodes. As in ENAS, we enable batch normalization in each node to prevent gradient explosion during architecture search, and disable it during architecture evaluation. Our recurrent network consists of only a single cell, i.e. we do not assume any repetitive patterns within the recurrent architecture.

Detailed experimental setup for this section can be found in Sect. A.1.2.

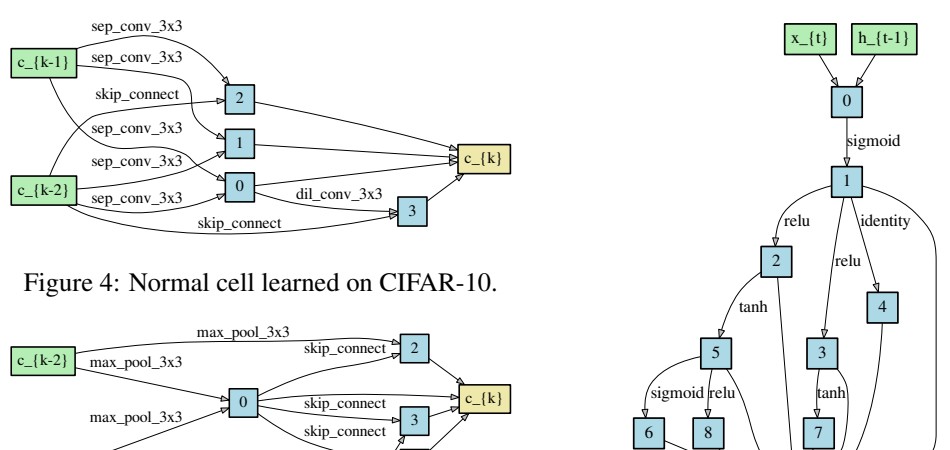

Figure 4: Normal cell learned on CIFAR-10.

Figure 5: Reduction cell learned on CIFAR-10.

Figure 6: Recurrent cell learned on PTB.

## 3.2 ARCHITECTURE EVALUATION

To determine the architecture for final evaluation, we run DARTS four times with different random seeds and pick the best cell based on its validation performance obtained by training from scratch for

a short period (100 epochs on CIFAR-10 and 300 epochs on PTB). This is particularly important for recurrent cells, as the optimization outcomes can be initialization-sensitive (Fig. 3).

To evaluate the selected architecture, we randomly initialize its weights (weights learned during the search process are discarded), train it from scratch, and report its performance on the test set. We note the test set is never used for architecture search or architecture selection.

Detailed experimental setup for architecture evaluation on CIFAR-10 and PTB can be found in Sect. A.2.1 and Sect. A.2.2, respectively. Besides CIFAR-10 and PTB, we further investigated the transferability of our best convolutional cell (searched on CIFAR-10) and recurrent cell (searched on PTB) by evaluating them on ImageNet (mobile setting) and WikiText-2, respectively. More details of the transfer learning experiments can be found in Sect. A.2.3 and Sect. A.2.4.

Table 1: Comparison with state-of-the-art image classifiers on CIFAR-10 (lower error rate is better). Note the search cost for DARTS does not include the selection cost (1 GPU day) or the final evaluation cost by training the selected architecture from scratch (1.5 GPU days).

| Architecture | Test Error (%) | Params (M) | Search Cost (GPU days) | #ops | Search Method |
|---|---|---|---|---|---|
| DenseNet-BC (Huang et al., 2017) | 3.46 | 25.6 | – | – | manual |
| NASNet-A + cutout (Zoph et al., 2018) | 2.65 | 3.3 | 2000 | 13 | RL |
| NASNet-A + cutout (Zoph et al., 2018)[†] | 2.83 | 3.1 | 2000 | 13 | RL |
| BlockQNN (Zhong et al., 2018) | 3.54 | 39.8 | 96 | 8 | RL |
| AmoebaNet-A (Real et al., 2018) | $3.34 \pm 0.06$ | 3.2 | 3150 | 19 | evolution |
| AmoebaNet-A + cutout (Real et al., 2018)[†] | 3.12 | 3.1 | 3150 | 19 | evolution |
| AmoebaNet-B + cutout (Real et al., 2018) | $2.55 \pm 0.05$ | 2.8 | 3150 | 19 | evolution |
| Hierarchical evolution (Liu et al., 2018b) | $3.75 \pm 0.12$ | 15.7 | 300 | 6 | evolution |
| PNAS (Liu et al., 2018a) | $3.41 \pm 0.09$ | 3.2 | 225 | 8 | SMBO |
| ENAS + cutout (Pham et al., 2018b) | 2.89 | 4.6 | 0.5 | 6 | RL |
| ENAS + cutout (Pham et al., 2018b)[*] | 2.91 | 4.2 | 4 | 6 | RL |
| Random search baseline[‡] + cutout | $3.29 \pm 0.15$ | 3.2 | 4 | 7 | random |
| DARTS (first order) + cutout | $3.00 \pm 0.14$ | 3.3 | 1.5 | 7 | gradient-based |
| DARTS (second order) + cutout | $2.76 \pm 0.09$ | 3.3 | 4 | 7 | gradient-based |

[*] Obtained by repeating ENAS for 8 times using the code publicly released by the authors. The cell for final evaluation is chosen according to the same selection protocol as for DARTS.
[†] Obtained by training the corresponding architectures using our setup.
[‡] Best architecture among 24 samples according to the validation error after 100 training epochs.

Table 2: Comparison with state-of-the-art language models on PTB (lower perplexity is better). Note the search cost for DARTS does not include the selection cost (1 GPU day) or the final evaluation cost by training the selected architecture from scratch (3 GPU days).

| Architecture | Perplexity | | Params (M) | Search Cost (GPU days) | #ops | Search Method |
|---|---|---|---|---|---|---|
| | valid | test | | | | |
| Variational RHN (Zilly et al., 2016) | 67.9 | 65.4 | 23 | – | – | manual |
| LSTM (Merity et al., 2018) | 60.7 | 58.8 | 24 | – | – | manual |
| LSTM + skip connections (Melis et al., 2018) | 60.9 | 58.3 | 24 | – | – | manual |
| LSTM + 15 softmax experts (Yang et al., 2018) | 58.1 | 56.0 | 22 | – | – | manual |
| NAS (Zoph & Le, 2017) | – | 64.0 | 25 | 1e4 CPU days | 4 | RL |
| ENAS (Pham et al., 2018b)[*] | 68.3 | 63.1 | 24 | 0.5 | 4 | RL |
| ENAS (Pham et al., 2018b)[†] | 60.8 | 58.6 | 24 | 0.5 | 4 | RL |
| Random search baseline[‡] | 61.8 | 59.4 | 23 | 2 | 4 | random |
| DARTS (first order) | 60.2 | 57.6 | 23 | 0.5 | 4 | gradient-based |
| DARTS (second order) | 58.1 | 55.7 | 23 | 1 | 4 | gradient-based |

[*] Obtained using the code (Pham et al., 2018a) publicly released by the authors.
[†] Obtained by training the corresponding architecture using our setup.
[‡] Best architecture among 8 samples according to the validation perplexity after 300 training epochs.

Table 3: Comparison with state-of-the-art image classifiers on ImageNet in the mobile setting.

| Architecture | Test Error (%) | | Params (M) | +× (M) | Search Cost (GPU days) | Search Method |
|---|---|---|---|---|---|---|
| | top-1 | top-5 | | | | |
| Inception-v1 (Szegedy et al., 2015) | 30.2 | 10.1 | 6.6 | 1448 | – | manual |
| MobileNet (Howard et al., 2017) | 29.4 | 10.5 | 4.2 | 569 | – | manual |
| ShuffleNet 2× ($g=3$) (Zhang et al., 2017) | 26.3 | – | ∼5 | 524 | – | manual |
| NASNet-A (Zoph et al., 2018) | 26.0 | 8.4 | 5.3 | 564 | 2000 | RL |
| NASNet-B (Zoph et al., 2018) | 27.2 | 8.7 | 5.3 | 488 | 2000 | RL |
| NASNet-C (Zoph et al., 2018) | 27.5 | 9.0 | 4.9 | 558 | 2000 | RL |
| AmoebaNet-A (Real et al., 2018) | 25.5 | 8.0 | 5.1 | 555 | 3150 | evolution |
| AmoebaNet-B (Real et al., 2018) | 26.0 | 8.5 | 5.3 | 555 | 3150 | evolution |
| AmoebaNet-C (Real et al., 2018) | 24.3 | 7.6 | 6.4 | 570 | 3150 | evolution |
| PNAS (Liu et al., 2018a) | 25.8 | 8.1 | 5.1 | 588 | ∼225 | SMBO |
| DARTS (searched on CIFAR-10) | 26.7 | 8.7 | 4.7 | 574 | 4 | gradient-based |

## 3.3 Results Analysis

The CIFAR-10 results for convolutional architectures are presented in Table 1. Notably, DARTS achieved comparable results with the state of the art (Zoph et al., 2018; Real et al., 2018) while using three orders of magnitude less computation resources (i.e. 1.5 or 4 GPU days vs 2000 GPU days for NASNet and 3150 GPU days for AmoebaNet). Moreover, with slightly longer search time, DARTS outperformed ENAS (Pham et al., 2018b) by discovering cells with comparable error rates but less parameters. The longer search time is due to the fact that we have repeated the search process four times for cell selection. This practice is less important for convolutional cells however, because the performance of discovered architectures does not strongly depend on initialization (Fig. 3).

**Alternative Optimization Strategies** To better understand the necessity of bilevel optimization, we investigated a simplistic search strategy, where $\alpha$ and $w$ are jointly optimized over the union of the training and validation sets using coordinate descent. The resulting best convolutional cell (out of 4 runs) yielded $4.16 \pm 0.16\%$ test error using 3.1M parameters, which is worse than random search. In the second experiment, we optimized $\alpha$ simultaneously with $w$ (without alteration) using SGD, again over all the data available (training + validation). The resulting best cell yielded $3.56 \pm 0.10\%$ test error using 3.0M parameters. We hypothesize that these heuristics would cause $\alpha$ (analogous to hyperparameters) to overfit the training data, leading to poor generalization. Note that $\alpha$ is not directly optimized on the training set in DARTS.

Table 2 presents the results for recurrent architectures on PTB, where a cell discovered by DARTS achieved the test perplexity of 55.7. This is on par with the state-of-the-art model enhanced by a mixture of softmaxes (Yang et al., 2018), and better than all the rest of the architectures that are either manually or automatically discovered. Note that our automatically searched cell outperforms the extensively tuned LSTM (Melis et al., 2018), demonstrating the importance of architecture search in addition to hyperparameter search. In terms of efficiency, the overall cost (4 runs in total) is within 1 GPU day, which is comparable to ENAS and significantly faster than NAS (Zoph & Le, 2017).

It is also interesting to note that random search is competitive for both convolutional and recurrent models, which reflects the importance of the search space design. Nevertheless, with comparable or less search cost, DARTS is able to significantly improve upon random search in both cases ($2.76 \pm 0.09$ vs $3.29 \pm 0.15$ on CIFAR-10; 55.7 vs 59.4 on PTB).

Results in Table 3 show that the cell learned on CIFAR-10 is indeed transferable to ImageNet. It is worth noticing that DARTS achieves competitive performance with the state-of-the-art RL method (Zoph et al., 2018) while using three orders of magnitude less computation resources.

Table 4 shows that the cell identified by DARTS transfers to WT2 better than ENAS, although the overall results are less strong than those presented in Table 2 for PTB. The weaker transferability between PTB and WT2 (as compared to that between CIFAR-10 and ImageNet) could be explained by the relatively small size of the source dataset (PTB) for architecture search. The issue of transferability could potentially be circumvented by directly optimizing the architecture on the task of interest.

Table 4: Comparison with state-of-the-art language models on WT2.

| Architecture | Perplexity | | Params (M) | Search Cost (GPU days) | Search Method |
|---|---|---|---|---|---|
| | valid | test | | | |
| LSTM + augmented loss (Inan et al., 2017) | 91.5 | 87.0 | 28 | – | manual |
| LSTM + continuous cache pointer (Grave et al., 2016) | – | 68.9 | – | – | manual |
| LSTM (Merity et al., 2018) | 69.1 | 66.0 | 33 | – | manual |
| LSTM + skip connections (Melis et al., 2018) | 69.1 | 65.9 | 24 | – | manual |
| LSTM + 15 softmax experts (Yang et al., 2018) | 66.0 | 63.3 | 33 | – | manual |
| ENAS (Pham et al., 2018b)[†] (searched on PTB) | 72.4 | 70.4 | 33 | 0.5 | RL |
| DARTS (searched on PTB) | 71.2 | 69.6 | 33 | 1 | gradient-based |

[†] Obtained by training the corresponding architecture using our setup.

## 4 CONCLUSION

We presented DARTS, a simple yet efficient architecture search algorithm for both convolutional and recurrent networks. By searching in a continuous space, DARTS is able to match or outperform the state-of-the-art non-differentiable architecture search methods on image classification and language modeling tasks with remarkable efficiency improvement by several orders of magnitude.

There are many interesting directions to improve DARTS further. For example, the current method may suffer from discrepancies between the continuous architecture encoding and the derived discrete architecture. This could be alleviated, e.g., by annealing the softmax temperature (with a suitable schedule) to enforce one-hot selection. It would also be interesting to explore performance-aware architecture derivation schemes based on the one-shot model learned during the search process.

## ACKNOWLEDGEMENTS

The authors would like to thank Zihang Dai, Hieu Pham and Zico Kolter for useful discussions.

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

# A EXPERIMENTAL DETAILS

## A.1 ARCHITECTURE SEARCH

### A.1.1 CIFAR-10

Since the architecture will be varying throughout the search process, we always use batch-specific statistics for batch normalization rather than the global moving average. Learnable affine parameters in all batch normalizations are disabled during the search process to avoid rescaling the outputs of the candidate operations.

To carry out architecture search, we hold out half of the CIFAR-10 training data as the validation set. A small network of 8 cells is trained using DARTS for 50 epochs, with batch size $64$ (for both the training and validation sets) and the initial number of channels 16. The numbers were chosen to ensure the network can fit into a single GPU. We use momentum SGD to optimize the weights $w$, with initial learning rate $\eta_w = 0.025$ (annealed down to zero following a cosine schedule without restart (Loshchilov & Hutter, 2016)), momentum 0.9, and weight decay $3 \times 10^{-4}$. We use zero initialization for architecture variables (the $\alpha$'s in both the normal and reduction cells), which implies equal amount of attention (after taking the softmax) over all possible ops. At the early stage this ensures weights in every candidate op to receive sufficient learning signal (more exploration). We use Adam (Kingma & Ba, 2014) as the optimizer for $\alpha$, with initial learning rate $\eta_\alpha = 3 \times 10^{-4}$, momentum $\beta = (0.5, 0.999)$ and weight decay $10^{-3}$. The search takes one day on a single GPU[3].

### A.1.2 PTB

For architecture search, both the embedding and the hidden sizes are set to 300. The linear transformation parameters across all incoming operations connected to the same node are shared (their shapes are all $300 \times 300$), as the algorithm always has the option to focus on one of the predecessors and mask away the others. Tying the weights leads to memory savings and faster computation, allowing us to train the continuous architecture using a single GPU. Learnable affine parameters in batch normalizations are disabled, as we did for convolutional cells. The network is then trained for 50 epochs using SGD without momentum, with learning rate $\eta_w = 20$, batch size 256, BPTT length 35, and weight decay $5 \times 10^{-7}$. We apply variational dropout (Gal & Ghahramani, 2016) of 0.2 to word embeddings, 0.75 to the cell input, and 0.25 to all the hidden nodes. A dropout of 0.75 is also applied to the output layer. Other training settings are identical to those in Merity et al. (2018); Yang et al. (2018). Similarly to the convolutional architectures, we use Adam for the optimization of $\alpha$ (initialized as zeros), with initial learning rate $\eta_\alpha = 3 \times 10^{-3}$, momentum $\beta = (0.9, 0.999)$ and weight decay $10^{-3}$. The search takes 6 hours on a single GPU.

## A.2 ARCHITECTURE EVALUATION

### A.2.1 CIFAR-10

A large network of 20 cells is trained for 600 epochs with batch size 96. The initial number of channels is increased from 16 to 36 to ensure our model size is comparable with other baselines in the literature (around 3M). Other hyperparameters remain the same as the ones used for architecture search. Following existing works (Pham et al., 2018b; Zoph et al., 2018; Liu et al., 2018a; Real et al., 2018), additional enhancements include cutout (DeVries & Taylor, 2017), path dropout of probability 0.2 and auxiliary towers with weight 0.4. The training takes 1.5 days on a single GPU with our implementation in PyTorch (Paszke et al., 2017). Since the CIFAR results are subject to high variance even with exactly the same setup (Liu et al., 2018b), we report the mean and standard deviation of 10 independent runs for our full model.

To avoid any discrepancy between different implementations or training settings (e.g. the batch sizes), we incorporated the NASNet-A cell (Zoph et al., 2018) and the AmoebaNet-A cell (Real et al., 2018) into our training framework and reported their results under the same settings as our cells.

---

[3]All of our experiments were performed using NVIDIA GTX 1080Ti GPUs.

### A.2.2 PTB

A single-layer recurrent network with the discovered cell is trained until convergence with batch size 64 using averaged SGD (Polyak & Juditsky, 1992) (ASGD), with learning rate $\eta_w = 20$ and weight decay $8 \times 10^{-7}$. To speedup, we start with SGD and trigger ASGD using the same protocol as in Yang et al. (2018); Merity et al. (2018). Both the embedding and the hidden sizes are set to 850 to ensure our model size is comparable with other baselines. The token-wise dropout on the embedding layer is set to 0.1. Other hyperparameters remain exactly the same as those for architecture search. For fair comparison, we do not finetune our model at the end of the optimization, nor do we use any additional enhancements such as dynamic evaluation (Krause et al., 2017) or continuous cache (Grave et al., 2016). The training takes 3 days on a single 1080Ti GPU with our PyTorch implementation. To account for implementation discrepancies, we also incorporated the ENAS cell (Pham et al., 2018b) into our codebase and trained their network under the same setup as our discovered cells.

### A.2.3 IMAGENET

We consider the *mobile* setting where the input image size is 224×224 and the number of multiply-add operations in the model is restricted to be less than 600M.

A network of 14 cells is trained for 250 epochs with batch size 128, weight decay $3 \times 10^{-5}$ and initial SGD learning rate 0.1 (decayed by a factor of 0.97 after each epoch). Other hyperparameters follow Zoph et al. (2018); Real et al. (2018); Liu et al. (2018a)[4]. The training takes 12 days on a single GPU.

### A.2.4 WIKITEXT-2

We use embedding and hidden sizes 700, weight decay $5 \times 10^{-7}$, and hidden-node variational dropout 0.15. Other hyperparameters remain the same as in our PTB experiments.

## B SEARCH WITH INCREASED DEPTH

To better understand the effect of depth for architecture search, we conducted architecture search on CIFAR-10 by increasing the number of cells in the stack from 8 to 20. The initial number of channels is reduced from 16 to 6 due to memory budget of a single GPU. All the other hyperparameters remain the same. The search cost doubles and the resulting cell achieves $2.88 \pm 0.09\%$ test error, which is slightly worse than $2.76 \pm 0.09\%$ obtained using a shallower network. This particular setup may have suffered from the enlarged discrepancy of the number of channels between architecture search and final evaluation. Moreover, searching with a deeper model might require different hyperparameters due to the increased number of layers to back-prop through.

## C COMPLEXITY ANALYSIS

In this section, we analyze the complexity of our search space for convolutional cells.

Each of our discretized cell allows $\prod_{k=1}^{4} \frac{(k+1)k}{2} \times (7^2) \approx 10^9$ possible DAGs without considering graph isomorphism (recall we have 7 non-zero ops, 2 input nodes, 4 intermediate nodes with 2 predecessors each). Since we are jointly learning both normal and reduction cells, the total number of architectures is approximately $(10^9)^2 = 10^{18}$. This is greater than the $5.6 \times 10^{14}$ of PNAS (Liu et al., 2018a) which learns only a single type of cell.

Also note that we retained the top-2 predecessors per node only at the very end, and our continuous search space before this final discretization step is even larger. Specifically, each relaxed cell (a fully connected graph) contains $2 + 3 + 4 + 5 = 14$ learnable edges, allowing $(7 + 1)^{14} \approx 4 \times 10^{12}$ possible configurations (+1 to include the *zero* op indicating a lack of connection). Again, since we are learning both normal and reduction cells, the total number of architectures covered by the continuous space before discretization is $(4 \times 10^{12})^2 \approx 10^{25}$.

---

[4]We did not conduct extensive hyperparameter tuning.

