# OpenReview forum: "DARTS: Differentiable Architecture Search"
_ICLR.cc/2019/Conference_

### Official Review · AnonReviewer3 · 2018-11-02
**Very interesting and promising approach**

**Rating:** 8
**Confidence:** 3

**Review:**

The authors introduce a continuous relaxation for categorical variables so as to utilize the gradient descent to optimize the connection weights and the network architecture. It is a cool idea and I enjoyed the paper.

One question, which I think is relevant in practice, is the initialization of the architecture parameters. I might be just missing, but I couldn't find description of the initial parameter values. As it is gradient based, it might be sensitive to the initial value of alpha.

In (5), the subscript for alpha should be removed as it defines a function of alpha. I think (5) is misleading as it is because of k-1. (and remove one "the" in "minimize the the validation" in the sentence above (5))

---

> ### Author Response · Authors · 2018-11-25
> **Response to AnonReviewer3**
>
> Thank you for the feedback.
>
> > Regarding the initialization of \alpha
> We use zero initialization which implies equal amount of attention (after taking the softmax) over all possible ops. At the early stage this ensures weights in every candidate op to receive sufficient learning signal (more exploration). This detail has been added to the revised draft.
>
> > “I think (5) is misleading as it is because of k-1.”
> Thank you for the suggestion. This has been fixed in the revised sect. 2.3.

---

### Official Review · AnonReviewer2 · 2018-11-02
**Very exciting; but also issues when you look into the details**

**Rating:** 7
**Confidence:** 5

**Review:**

This paper proposes a novel way to formulate neural architecture search as a differentiable problem.
It uses the idea of weight sharing introduced in previous papers (convolutional neural fabrics, ENAS, and Bender et al's one shot model) and combines this with a relaxation of discrete choices between k operators into k continuous weights. Then, it uses methods based on hyperparameter gradient search methods to optimize in this space and in the end removes the relaxation by dropping weak connections and selecting the single choice of the k options with the highest weight. This leads to an efficient solution for architecture search. Overall, this is a very interesting paper that has already created quite a buzz due to the simplicity of the methods and the strong results. It is a huge plus that there is code with the paper! This will dramatically increase the paper's impact.
In my first read through, I thought this might be a candidate for an award paper, but the more time I spent with it the more issues I found. I still think the paper should be accepted, but I do have several points of criticism / questions I detail below, and to which I would appreciate a response.

Some criticisms / questions:

1. The last step of how to move from the one-shot model to a single model is in a sense the most interesting aspect of this work, but also the one that leaves the most questions open: Why does this work? Are there cases where we lose arbitrarily badly by rounding the solution to the closest discrete value or is the performance loss bounded? How would other ways of moving from the relaxation to a discrete choice work? I don't expect the paper to answer all of these questions, but it would be useful if the authors acknowledge that this is a critical part of the work that deserves further study. Any insights from other approaches the authors may have tried before the mechanism in Section 2.4 would also be useful.

2. The related work is missing several papers, namely the entire category of work on using network morphisms to speed up the optimization process, Bender et al's one shot model, and several early papers on neural architecture search (work on NAS did not only start in 2017 but goes back to work in the 1990s on neuroevolution that is very similar to the evolution approach by Real). This is a useful survey useful for further references: https://arxiv.org/abs/1808.05377

3. I find a few of the claims to be a bit too strong. In the introduction, the paper claims to outperform ENAS, but really the paper doesn't give a head-to-head comparison. In the experiments, ENAS is faster and gives slightly worse results. The authors state explicitly that their method is slower because they run it 4 times and pick the best result. One could obviously also do that with ENAS, and since ENAS is 8 times faster one could even run it 8 times! This is unfair and should be fixed. I don't really care even if it turns out that ENAS performs a bit better with the same budget, but comparisons should be fair and on even ground in order to help our science advance -- something that is far too often ignored in the ML literature in order to obtain a table with bold numbers in one's own row.
Likewise, why is ENAS missing in the Figure 3 plots for CIFAR, and why is its performance not plotted over time like that of DARTS?

4. The paper is not really forthcoming about clearly stating the time required to obtain the results:
- On CIFAR, there are 4 DARTS run of 1 day each
- Then, the result of each of these is evaluated for 100 epochs (which is only stated in the caption of Figure 3) to pick the best. Each of these validation runs takes 4 hours (which, again, one has to be inferred from the fact that random search can do 24 such evaluations in 4 GPU days), so this step takes another 16 GPU hours.
- Then, one needs to train the final network for 600 epochs; this is a larger network, so this should take another 2-3 GPU days.
So, overall, to obtain the result on CIFAR-10 requires about one GPU week. That's still cheap, but it's a different story than 1 day.
Likewise, DARTS is *not* able to obtain 55.7 perplexity on PTB in 6 hours with 4 GPUs; again, there is the selection step (probably another 4*6 hours?) and I think training the final model takes about 2 GPU days. These numbers should be stated prominently next to the stated "search times" to not mislead the reader.

5. One big question I have is where the hyperparameters come from, for both the training pipeline and the final evaluation pipeline (which actually differ a lot!).
For example, here are the hyperparameters for CIFAR, in this format: training pipeline value -> final evaluation pipeline value:
#cells: 8 -> 20
batch size: 64 -> 96
initial channels: 16 -> 36
#epochs: 50 -> 600
droppath: no -> yes (with probability 0.2)
auxiliary head: no -> yes (with weight 0.4)
BatchNorm: enabled (no learnable parameters) -> enabled

The situation is similar for PTB:
embedding size: 300 -> 850
hidden units per RNN layer: 300 -> 850
#epochs: 500 -> 8000
batch size: 256 (SGD) -> 64 (ASGD), sped up by starting with SGD
weight decay: 5e-7 -> 8e-7
BatchNorm: enabled (no learnable parameters) -> disabled

The fact that there are so many differences in the pipelines is disconcerting, since it looks like a lot of manual work is required to get these right. Now you need to tune hyperparameters for both the training and the final evaluation pipeline? If you have to tune them for the final evaluation pipeline, then you can't capitalize at all on the fact that DARTS is fast, since hyperparameter optimization on the full final evaluation pipeline will be order of magnitudes more expensive than running DARTS.

6. How was the final evaluation pipeline chosen? Before running DARTS the first time, or was it chosen to be tuned for architectures found by DARTS?

7. A question about how the best of 4 DARTS runs is selected, and how the best of the 24 random samples in random search is evaluated: is this based on 100 epochs using the *training* procedure or the *final evaluation* procedure? Seeing how different the hyperparameters are above, this should be stated.

8. A few questions to the authors related to the above: how did you choose the hyperparameters of DARTS? The DARTS learning rate for PTB is 10 times higher than for CIFAR-10, and the momentum also differs a lot (0.9 vs. 0.5). Did you ever consider different hyperparameters for DARTS? If so, how did you decide on the ones used? Is it sensitive to the choice of hyperparameters? In the author response period, could you please report the
(1) result of running DARTS on PTB using the same DARTS hyperparameters as used for CIFAR-10 (learning rate 3*e-4 and momentum (0.5,0.999)) and
(2) result of running DARTS on CIFAR-10 using the same DARTS hyperparameters as used for PTB (learning rate 3*e-3 and momentum (0.9,0.999))?

9. DARTS is being critizized in https://openreview.net/pdf?id=rylqooRqK7#page=10&zoom=180,-16,84
I am wondering whether the authors have a reply to this.
The algorithm for solving the relaxed problem is also not mathematically derived from the optimization problem to be solved (equations 3,4), but it is more a heuristic. A derivation, or at least a clearer motivation for the algorithm would be useful.

10. Further comments:
- Equation 1: This looks like a typo, shouldn't this be x(j) = \sum_{i<j} o(i,j) x(i) ? Even if the authors wanted to use the non-intuitive way of edges going from j to i, then o(i,j) should still be o(j,i).
- Just above Equation 5: "the the"
- Equation 5: I would have found it more intuitive had \alpha_{k-1} already just been a generic \alpha here.
- It would be nice if the authors gave the explicit equations for the extension with momentum in the appendix for completeness.
- The authors should include citations for techniques such as batch normalization, Adam, and cosine annealing.


Despite these issues (which I hope the authors will address in the author response and the final version), as stated above, I'm arguing for accepting the paper, due to the simplicity of the method combined with its very promising results and the direct availability of code.

---

> ### Author Response · Authors · 2018-11-25
> **Response to AnonReviewer2 (1/2)**
>
> Thank you for the detailed comments and questions. We have fixed the missing references (Q2) and presentational issues (Q4, Q10) in the revision. Below we focus on the major points:
>
> > Regarding discretization schemes (Q1)
> The current discretization scheme can be viewed as a heuristic to minimize the per-node rounding error, as described in the revised sect. 2.4. While refining this part was not our primary focus, it indeed deserves further study. We have also added a remark in the draft to make readers aware of this potential limitation.
>
> To reduce the rounding error, in our preliminary experiments we tried annealing the softmax temperature to enforce one-hot selection, but did not observe clear differences in terms of the quality of the derived cells. Note that a large rounding error does not necessarily imply poor performance, since the current discretization mechanism only depends on the ranking among the strengths of the incoming edges.
>
> > “since ENAS is 8 times faster one could even run it 8 times” (Q3)
> We agree it would be informative to compare DARTS and ENAS given the same search cost (e.g., 4 GPU days). Following your suggestion, we repeated the search process of ENAS for 8 times on CIFAR-10 using the authors' implementation and their best setup. We then used the same selection protocol as for DARTS by training the candidate cells for 100 epochs using half of the CIFAR-10 training data to get the validation performance on the other half. The best ENAS cell out of 8 runs achieves 2.91% test error using 4.2M params in the final evaluation, which is slightly worse than 4 runs of DARTS (2.76% error using 3.3M params). These new results have been included in Table 1 of the revised draft.
>
> > “One big question I have is where the hyperparameters come from” (Q5, Q6).
> Let us explain our reasoning for each of these hyperparameters in detail:
>
> For convolutional cells:
>
> Our setup of #cells (8->20), #epochs (600) and weight for the auxiliary head (0.4) in the final evaluation exactly follows Zoph et al., 2018. The #init_channels is enlarged from 16 to 36 to ensure a comparable model size (~3M) with other baselines. Given those settings, we then use the largest possible batch size (96) for a single GPU. The drop path probability was tuned wrt the validation set among the choices of (0.1, 0.2, 0.3) given the best cell learned by DARTS.
>
> We treat droppath, auxiliary towers and cutout as additional augmentations only for the final evaluation. Learnable affine parameters in the batch normalisation are disabled during the search phase to avoid arbitrary rescaling of the nodes, as explained in sect A.1.1. They are enabled in the evaluation phase to ensure fair comparison with other baseline networks.
>
> For recurrent cells:
>
> We always use the same #units for both embedding and hidden layers, which is enlarged from 300 to 850 in the final evaluation to make our #params (~23M) comparable with other models in the literature. We then use the largest possible batch size (64) to fit our model in a single GPU. The l2 weight decay was tuned on the validation set given the best recurrent cell. We do not trigger ASGD during the search phase for simplicity and also to accommodate our current approximation scheme which does not take into account model averaging (though it can be modified to support it).
>
> Batch normalisation is useful during architecture search to prevent gradient explosion (Sect 3.1.2). Similar to the case of convnets, learnable affine params are disabled to avoid node rescaling, as explained in A.1.1 and A.1.2. Once the cell is learned, batch normalisation layers are omitted in the final evaluation for fair comparison with existing language models which usually do not involve normalisation. Our usage of batch normalisation for RNN architecture search follows ENAS.
>
> > “how the best of the 24 random samples in random search is evaluated” (Q7):
> The same script is used for cell selection of DARTS and random search. All the hyperparameters, except #epochs, are identical to those in our final evaluation pipeline.

---

> > ### Author Response · Authors · 2018-11-25
> > **Response to AnonReviewer2 (2/2)**
> >
> > > “how did you choose the hyperparameters of DARTS” (Q8)
> > While Adam with a small learning rate (3e-4) and the default first momentum 0.9 works well for recurrent cells, the same setup leads to slow progress for conv cells (\alpha would remain near-uniform in 50 epochs). We thus (1) increased the learning rate by an order of magnitude to 3e-3 and (2) lowered the momentum from 0.9 to 0.5 to alleviate instability due to the increased learning rate.
> >
> > To better understand the effect of different momentums, we have now repeated our CIFAR-10 experiments using momentum 0.9 instead. The newly obtained cells achieve 2.89% test error with 3.5M params (1st order) and 2.91% with 3.3M params (2nd order). These are comparable with our previous results based on momentum 0.5.
> >
> > > “I am wondering whether the authors have a reply to this” (Q9)
> > In DARTS we use a deterministic architecture encoding, where \alpha is a continuous variable with well-defined gradients. While being conceptually simple, the method may suffer from bias due the discrepancy between \alpha and the derived discrete architecture.
> >
> > The key idea of SNAS is to replace the deterministic encoding in DARTS with a stochastic one. This modification makes architecture derivation more straightforward as \alpha is now a discrete random variable by definition. Unlike DARTS, gradients wrt (the distribution of) \alpha are no longer well-defined, hence Gumbel-softmax estimator is used to enable a differentiable optimization procedure. As a result, the estimated gradients are biased as long as the temperature is not zero.
> >
> > As far as the empirical results are concerned, the two methods perform similarly on CIFAR-10, though the DARTS cell transfers slightly better to ImageNet. The ability of DARTS to learn the architectures of recurrent cells has also been empirically verified by its strong performance for language modeling (Table 2), whereas that of SNAS requires future investigation.
> >
> > > “A derivation, or at least a clearer motivation for the algorithm would be useful.” (2nd part of Q9)
> > Please refer to our response to AnonReviewer1 and our revised sect. 2.3.

---

### Official Review · AnonReviewer1 · 2018-11-02
**Well exposed incremental improvement to architechture tuning that gives state-of-the-art models on two classic (but old) benchmarks**

**Rating:** 6
**Confidence:** 2

**Review:**

(Disclaimers: I am not not active in the sub-field, just generally interested in the topic, it is easy however to find this paper in the wild and references to it, so I accidentally found out the name of the authors, but had not heard about them before reviewing this, so I do not think this biased my review).

DARTS, the algorithm described in this paper, is part of the one-shot family of architecture search algorithms. In practice this means training an over-parameterized architecture is, of which the architectures being searched for are sub-graphs. Once this bigger network is trained it is pruned into the desired sub-graph. DARTS has "indicator" weights that indicate how active components are during training, and then alternatively trains these weights (using the validation sets), and all other weights (using the training set). Those indicators are then chosen to select the final sub-graph.

More detailed comments:

It seems that the justification of equations (3) and (4) is not immediately obvious, in particular, from an abstract point of view, splitting the weights into w, and \eta to perform the bi-level optimizations appears somewhat arbitrary. It almost looks like optimizing the second over the validation could be interpreted as some form of regularization. Is there a stronger motivation than that is similar to more classical model/architecture selection?

There are some papers that seem to be pretty relevant and are worth looking at and that are not in the references:

http://proceedings.mlr.press/v80/bender18a.html
https://openreview.net/forum?id=HylVB3AqYm (under parallel review at ICLR, WARNIGN TO REVIEWERS: contains references to a non anonymized version of this paper )

I think architecture pruning literature is relevant too, it would be nice to discuss the connection between NAS and this sub-field, as I think there are very strong similarity between the two.

Pros:
* available source code
* good experimental results
* easy to read
* interesting idea of encoding how active the various possible operations are with special weights

Cons
* tested on a limited amount of settings, for something that claims that helps to automate the creation of architecture, in particular it was tested on two data set on which they train DARTS models, which they then show to transfer to two other data sets, respectively
* shared with most NAS papers: does not really find novel architectures in a broad sense, instead only looks for variations of a fairly limited class of architectures
* theoretically not very strong, the derivation of the bi-level optimization is interesting, but I believe it is not that clear why iterating between test and validation set is the right thing to do, although admittedly it leads to good results in the settings tested

---

> ### Author Response · Authors · 2018-11-25
> **Response to AnonReviewer1**
>
> Thank you for the feedback.
>
> > “It seems that the justification of equations (3) and (4) is not immediately obvious”
> In this work we treat \alpha as a high-dimensional hyperparameter. The bilevel formulation offers a mathematical characterization of the standard hyperparameter tuning procedure, namely to find hyperparameter \alpha that leads to the best validation performance (eq. (4)) after regular parameters w are trained until convergence on the training set (eq. (3)) given \alpha.
>
> > "it is not that clear why iterating between test and validation set is the right thing to do"
> Using two separate data splits for \alpha and w as in the bilevel formulation should effectively prevent hyperparameter/architecture from overfitting the training data. Advantage of doing so has also been empirically verified by our experiments. Please refer to “Alternative Optimization Strategies” in sect. 3.3 of the revised draft.
>
> From the algorithmic point of view, each architecture gradient step consists of two subroutines:
> (i) Obtaining w^*(\alpha), namely weights trained until convergence for the given architecture, by solving the inner optimization eq (4). This can normally be achieved by taking a large number of gradient descent steps of w wrt the training loss.
> (ii) Descending \alpha wrt the validation loss defined based on w^*(\alpha).
> Our iterative algorithm is a truncated version of the above by approximating the optimization procedure in (i) using only a single gradient step.
>
> > “I think architecture pruning literature is relevant too”
> Yes, network pruning and (differentiable) architecture search are related despite somewhat different goals. The former aims to learn fine-grained sparsity patterns (e.g. which neurons or channels should be kept) that best approximate a given unpruned network. The latter aims to learn macro-level sparsity patterns that represent an architecture.

---

### Public Comment · (anonymous) · 2018-10-03
**interesting works**

Hello there,

It is apparently an interesting work with solid results on a variety of dataset.

I have a quick question, the paper tries to model the architecture design domain as a function, then the agent searches for the promising architectures with the gradient descent.

So, what's the key difference between the surrogate function in Progressive Neural Architecture Search? The surrogate model is also differentiable, and the idea, in my perspective, would be similar.

Also the simulation model in "AlphaX: eXploring Neural Architectures with Deep Neural Networks and Monte Carlo Tree Search" is also differentiable, and potentially to achieve the same goal.

Could you please clarify these points? Thank you.

---

> ### Author Response · Authors · 2018-10-03
> **Thank you for the comments.**
>
> SMBO (used in PNAS) and MCTS are discrete search algorithms. Both do not offer an explicit notion of gradient over (the continuous representation of) the architecture as in DARTS.
>
> The goal of the performance predictor/surrogate model in SMBO is to guide the search within the discrete space. This alone does not make the search algorithm itself differentiable.

---

### Public Comment · (anonymous) · 2018-10-03
**Questions about the comparison.**

This is an interesting work with awesome codes. I have a few questions about the experimental comparison.

1. This paper uses a different search space than NAS/PNAS/ENAS, i.e., 8 different operations with only 4 steps. Is it unfair to compare the search cost with those methods? For example, NAS uses 13 operators and tries 2 connections, the search space is much larger than DARTS. Would it be better to use the same search space for comparison?

2. Why use dilated convolution? Most previous NAS works seem not to use dilated convolutions.

3. Would you mind to discuss the effect of the network depth during searching? In A.1.1, the network with 8 cells is used to search the best cell. I try the released code and use a deeper network (20 cells) for searching, but obtain much worse results than DARTS. Is there any explanation?

---

> ### Author Response · Authors · 2018-10-03
> **Clarifications.**
>
> Thank you for the comments.
>
> >> Regarding the number of operations
> The #ops in our convnet experiments is the same (eight) as in PNAS [1] and is greater than 6 used in ENAS [2]. We didn’t try larger numbers due to the memory constraints of a single GPU. We will include the #ops as a column in our Tables to better reflect these details.
>
> >> "the search space is much larger than DARTS"
> This is not correct. While the controller in NAS must sample exactly 2 connections per node, DARTS is simultaneously exploring all possible connections within a fully-connected supergraph. Although we kept the top-k (k=2) connections in the derived discrete architecture (sect. 2.4) for fair comparison with NAS, with DARTS k could be other numbers greater than 2.
>
> >> "Most previous NAS works seem not to use dilated convolutions."
> This is not correct. Dilated convolutions are used in most prior works. Please refer to NASNets [3], AmoebaNets [4] and PNASNets [1].
>
> >> "Would you mind to discuss the effect of the network depth during searching?"
> Since \alpha is shared among cells at different layers, backprop wrt \alpha behaves similarly to BPTT. Searching with a deeper network might thus require different hyper-parameters due to the increased number of layers (steps) to back-prop through.
>
> [1] Liu, Chenxi, et al. "Progressive neural architecture search." arXiv preprint arXiv:1712.00559 (2017).
> [2] Pham, Hieu, et al. "Efficient Neural Architecture Search via Parameter Sharing." arXiv preprint arXiv:1802.03268 (2018).
> [3] Zoph, Barret, et al. "Learning transferable architectures for scalable image recognition." arXiv preprint arXiv:1707.070122.6 (2017).
> [4] Real, Esteban, et al. "Regularized evolution for image classifier architecture search." arXiv preprint arXiv:1802.01548(2018).

---

> > ### Public Comment · (anonymous) · 2018-10-09
> > **Thanks for your kind reply.**
> >
> > Thanks for your kind reply.

---

> > > ### Author Response · Authors · 2018-10-14
> > > **Additional results using 20 cells**
> > >
> > > You are welcome. We also conducted architecture search using 20 cells (with initial #channels reduced from 16 to 6 due to memory budget) without adjusting other hyperparameters. The resulting cell achieved 2.88 +/- 0.09% test error on CIFAR-10. We will include those additional results and related discussion in the revised paper.

---

> > > > ### Public Comment · (anonymous) · 2018-10-27
> > > > **First order or second order?**
> > > >
> > > > Does the "2.88 +/- 0.09%" come from DARTS (first order) or DARTS (second order)?
> > > >
> > > > In addition, would you mind to report the results of DARTS (first order) on WT2?

---

### Public Comment · (anonymous) · 2018-10-15
**CIFAR-10 test set as validation set for architecture evaluation**

Dear authors,

I noticed that in your architecture evaluation script (cnn/train.py) for CIFAR-10, you use the whole training set of 50k images for training and declare the test set as your validation set. To my knowledge, this is not common practice and will result in a lower test error compared to others who split the training set into 45k/5k train/validation (as, for example, in the Resnet and Densenet papers), while evaluating on the test set only once at the very end of the training procedure.

I suggest you rerun your experiments with a 45k/5k train/validation split to ensure a fair comparison, or please clarify if there is a misunderstanding.

Thank you.

---

> ### Author Response · Authors · 2018-10-15
> **Test set was never used as validation set**
>
> First, we'd like to emphasize that the hyperparameters provided in our scripts were chosen based on a random subset of the training data (as the validation set) rather than the test data, though we used the 50K/10K training/test split in our released code (i.e., cnn/train.py for the final run) and printed out the errors on both sets. This is to make it easier for people to reproduce the expected test learning curves and the reported test error of *the model at the very end of training*.
>
> Secondly, we'd like to point out that training the final model using all the 50K images to obtain the test error on the 10K images is a common practice. Please refer to ResNet [1] (Sect. 4.2), DenseNet [2] (Sect. 4.1: “For the final run we use all 50,000 training images and report the final test error at the end of training”), their official implementations, as well as the codebases of NAS and ENAS. Note the 45K/5K split is recommended for model selection (architecture search and hyperparameter tuning) but not for the final run.
>
> Finally, we agree that this is an important detail that should be included in the paper. We also plan to refactor our code to ensure the users do not mistakenly tune their models wrt the test set. Thanks for bringing it up and please let us know if you have any other concerns.
>
> [1] He, Kaiming, et al. "Deep residual learning for image recognition." Proceedings of the IEEE conference on computer vision and pattern recognition. 2016.
> [2] Huang, Gao, et al. "Densely Connected Convolutional Networks." CVPR. Vol. 1. No. 2. 2017.

---

### Public Comment · (anonymous) · 2018-10-17
**Reference Missing & Questions about formula 5**

Hi there,

This is absolutely a good work, however, there might be some small questions.

Firstly, there is a paper about NAS not mentioned, accepted by CVPR 2018. By using Q-Learning, it achieves comparable results on ImageNet within 3 days on 32 GPU. It might be better to mention and add comparison with this work. The link is here https://arxiv.org/abs/1708.05552.

Secondly, the algorithm 1 with the formula 5 seems a little bit confusing. Would it be more clear and distinguishable to give complete expression or formula about w_k, w_k-1, and w_prime in algorithm 1?

Lastly, I am kind of curious about the motivation of formula 5, could you give more detailed demonstration or experiment results about the comparison between the vanilla GD and current formula 5?

---

> ### Author Response · Authors · 2018-10-18
> **Thank you for the questions**
>
> > “there is a paper about NAS not mentioned”
> Thanks for mentioning about the BlockQNN paper. We will cite it as a method under the RL category.
>
> > Regarding definitions of w’ and w_k
> w’ means the one-step unrolled w, whose definition is given underneath eq (6). w_k means the actual numerical value of w at step k. We’ll make these more clear in the revision.
>
> > “I am kind of curious about the motivation of formula 5”
> Please refer to section 2.3. The motivation is to descent the architecture wrt the optimal w* instead of the current suboptimal w. The former is expensive but can be approximated by the latter after taking a gradient step. While the idea of unrolling is new to the NAS literature, similar techniques can be found in unrolled GAN [1] and MAML [2].
>
> > “the comparison between the vanilla GD and current formula 5?”
> We do have provided results to compare formula 5 (DARTS 2nd order in Table 1 & 2), vanilla GD (DARTS 1st order in Table 1 & 2) and coordinate descent (2nd paragraph in section 3.3).
>
> [1] Metz, Luke, et al. "Unrolled generative adversarial networks." arXiv preprint arXiv:1611.02163 (2016).
> [1] Finn, Chelsea, Pieter Abbeel, and Sergey Levine. "Model-agnostic meta-learning for fast adaptation of deep networks." arXiv preprint arXiv:1703.03400 (2017).

---

> > ### Public Comment · (anonymous) · 2018-10-19
> > **Thank you for your kind reply**
> >
> > Thank you for your kind reply, it is indeed a very good paper worth reading and reflecting.

---

### Public Comment · (anonymous) · 2018-10-20
**Complexity of this search space**

Hi there,

In ENAS, they show that their search space can realize  1.3*10^11 final networks in section 2.4, and that of PNAS is ~10^12 as calculated in section 3.1, what is the complexity of the search space in this paper? Could you add a table to compare the complexity of search space in these papers, or add a column in table 1&2 to show the efficiency i.e networks searched per GPU hour? It seems to be more convincing if the comparison of complexity of search space could be provided.

---

> ### Author Response · Authors · 2018-10-21
> **Complexity analysis**
>
> Thanks for the question. We will include complexity analysis in the revised paper.
>
> As for ConvNets, each of our discretized cell allows \prod_{k=1}^4 ((k+1)*k)/2)*(7^2) = ~10^9 possible DAGs (recall we have 7 non-zero ops, 2 input nodes, 4 intermediate nodes with 2 predecessors each) without considering graph isomorphism. Since we jointly learn both normal and reduction cells, the total #architectures is approximately (10^9)^2 = 10^18. This is greater than the ~5.6*10^14 of PNAS (reported in their sect 3.1) which learns only a single type of cell.
>
> Also note that we retained the top-2 predecessors per node only in the very end, and our continuous search space before this final discretization step is even larger. Specifically, each relaxed cell (a fully connected graph) contains 2+3+4+5 = 14 learnable edges, allowing (7+1)^14 = ~4*10^12 possible configurations (+1 to include the zero op indicating a lack of connection). Again, since we are learning both normal and reduction cells, the total number of architectures covered by the continuous space before discretization is (4*10^12)^2 = ~10^25. The above assumes that we retain only 1 of the 8 ops per edge, as done in our experiments. The search space can be substantially enlarged without additional computation overhead by retaining multiple ops per edge (e.g. by replacing the current argmax during discretization with top-K selection). We leave the exploration of this enriched space as our future work.
>
> > "networks searched per GPU hour?"
> This metric is not directly applicable to DARTS, which optimizes architectures in continuous space in contrast to most prior works that enumerate architecture samples.

---

> > ### Public Comment · (anonymous) · 2018-10-22
> > **Thanks for your kind reply**
> >
> > Cool!!
> > Thanks for your kind reply

---

### Public Comment · (anonymous) · 2018-10-22
**Maybe some other baselines should also be reported**

Hi,

I have some questions about the optimization of DARTS.

1. In Equation 3 and 4, the w*(alpha) is obtained from the training set and alpha is optimized on validation set. I know it makes sense to use validation set for alpha, as is discussed in the paper. However, I wonder what the performance will be if you optimize w and alpha both on training set? If you split the training set to a "train" and a "valid" set half and half, as you did in the code, the generalization would be better, but less samples is used to train alpha and w. However, if you use the whole training set, more samples are seen by the model to optimize alpha and w, and the performance might also be better.  In my opinion, this should also be a baseline for completeness.

2. This question is associated with the above one. In algorithm 1, the alpha and w are optimized alternatively. My question is: If   w and alpha are both optimized on the same training set, can we optimize alpha and w simultaneously without alternating?

In summary, I think it would be better to provide results on using the same set to optimize w and alpha, and also compare the alternating update manner with the simultaneous updating when the same set is used to optimize w and alpha.

Thanks

---

> ### Author Response · Authors · 2018-10-24
> **Please refer to sect 3.3**
>
> Thank you for the questions.
>
> > "it would be better to provide results on using the same set to optimize w and alpha"
> The results using this strategy are already presented in the 2nd paragraph of sect 3.3. The corresponding cell yielded 4.16 ± 0.16% test error.
>
> > "also compare the alternating update manner with the simultaneous updating"
> Following your suggestion, we further treated \alpha as part of conventional parameters and optimized it simultaneously with w. The resulting cell yielded 3.56 ± 0.10% test error.
>
> To summarize, both schemes are worse than the original bilevel formulation (2.76 ± 0.09% test error), which we attribute to overfitting — note \alpha is "tuned" directly on the training set in the suggested heuristics. We will expand our sect 3.3 to include more discussions.

---

### Public Comment · (anonymous) · 2018-10-25
**it seems the way to derive the discrete architectures is wired**

The way to derive the final discrete architectures is to select the k strongest predecessors according to Sec. 2.4. However, it seems it is not consistent with the training objective. Specifically, in training, the model is optimized in the condition that all possible ops for each edge are summarized according to the weights by the softmax of alpha. Selecting the k strongest predecessors to derive the final architecture cannot ensure the discrete one is the best.  Actually, the "quantization error" might leads the final architecture to be totally different with the one from the training procedure.

I have run the code, and I also find the alphas seems quite wired: Most of them have max value on the same op. This is also confirm by the figure 4 and 5 of the paper, i.e. ALL of ops for reduce cell is max pooling and most of ops for normal cell is sep_conv_3x3. It is really wired.

Can you prove that the discrete one is the best architecture? And Could you provide the values of  alphas for the normal and reduce cell for figure 4 and 5?

---

> ### Author Response · Authors · 2018-10-25
> **Better discretization is possible but orthogonal to our focus**
>
> Thank you for the comments.
>
> > “Selecting the k strongest predecessors to derive the final architecture cannot ensure the discrete one is the best”
> We retained 2 predecessors per node in order to make our derived cells comparable with the ones in prior works (NAS/PNAS/ENAS/AmoebaNets). This is for fair comparison but by no means the optimal discretization strategy.
>
> > “Actually, the "quantization error" might leads the final architecture to be totally different with the one from the training procedure.”
> It’s expected that continuous relaxation would come with a tradeoff between efficiency and bias. Quantization error of such kind can be reduced, e.g., by annealing the softmax temperature throughout the search process, forcing the \alpha’s to approach one-hot vectors. Improving our current discretization strategy at the end of search is an interesting direction orthogonal to our main focus, i.e. the overall framework of differentiable architecture search.
>
> > “ALL of ops for reduce cell is max pooling and most of ops for normal cell is sep_conv_3x3”
> First, this is incorrect. Our learned reduction cell contains not only max pooling but also skip connections; our learned normal cell contains not only sep_conv_3x3, but also skip connections and dilated convs. Please refer to Figure 4 & 5.
>
> Secondly, it’s actually interesting that the algorithm learns to introduce more translation invariance in the reduction cell (through multiple pooling ops) and to come up with a densely connected normal cell (through 3x3 sep convs and skip connections). Both design patterns are existent in successful architectures designed by human experts.
>
> > “It is really wired.”
> While visual judgements about cells in Figure 4 & 5 can be subjective, please note (1) effectiveness of those cells has been quantitatively verified by their competitive performance on both CIFAR-10 and ImageNet; (2) the algorithm can learn to leverage a more diverse set of ops when necessary. Please refer to our recurrent cell in Figure 6 with strong results on PTB.

---

> > ### Public Comment · (anonymous) · 2018-11-01
> > **On the ZERO operation**
> >
> > I have tried your suggestion on annealing the softmax to see the effect of discretization, and there seems to be little difference in the derived network. However, put it aside, when I inspect into the derivation method provided in your implementation, I find that the operation ZERO is omitted (the final operation is selected from any ops but ZERO), as in your code darts/cnn/model_search.py:146. Then I go back to check the logit of ZERO operation, and find it is actually the largest in almost every edge of the normal cell.
> >
> > To exclude the effect of annealing, I run your original implementation for three times with different random seeds. And it seems ZERO is still the one with largest logit. If the ZERO operation is playing the role as you stated in Sec. 2.1, it should be the argmax that is supposed to be selected as you stated in Sec. 2.4, resulting in an extremely sparse graph rather than the one you provided. Could you please give an explanation to
> > 1) why ZERO operation is omitted in both edge and operation selection?
> > 2) why ZERO operation tends to have largest logit?

---

> > > ### Author Response · Authors · 2018-11-02
> > > **Clarification regarding zero ops**
> > >
> > > Thank you for the questions.
> > >
> > > > "why ZERO operation is omitted in both edge and operation selection?"
> > > We’d like to point out that the zero op does play a role in determining the predecessors for each node (edge selection). Please refer to the edge strength defined in sect 2.4.
> > >
> > > Once the predecessors are determined, the zero operations are no longer used in argmax (op selection) for two reasons:
> > > (1) To make our derived networks comparable with NAS/PNAS/ENAS/AmoebaNets, which all assume a fixed sparsity level, i.e., exactly two predecessors per node via *non-zero* ops.
> > > (2) The strengths of zero ops can be underdetermined, as will be explained below.
> > >
> > > > "why ZERO operation tends to have largest logit?"
> > > Note the behavior of the network is not sensitive to the output scale of the mixed ops due to the presence of batchnorm. This makes the strength of the zero operation underdetermined, because we can always add some incremental value to the logit of a zero op (which is equivalent to rescaling the mixed op it belongs to, according to eq (2) in sect. 2.2) with a little effect on the final classification outcome.
> > >
> > > The above is not an issue with our current discretization scheme, which is based on the relative importance among non-zero ops only (once the active predecessors are decided). We will add more discussions on this topic in the revised paper.

---

> > > > ### Public Comment · (anonymous) · 2018-11-02
> > > > **Follow-up on ZERO ops**
> > > >
> > > > Thank you for your reply.
> > > >
> > > > 1) The claim that ZERO is omitted in edge selection is based on my understanding of your code at darts/cnn/search_model.pg:142. But I am not sure whether I comprehend it correctly. Could you please give an explanation?
> > > >
> > > > 2) The explanation that ZERO does not play a role in the relative value of a feature map makes a lot of sense. But it also puts lots of weights on the fact that the mix op is continuous. If the softmax is annealed, this mixing effect is supposed to be diminishing. Rather than ZERO, a truly effective operation should comes up as max. However, in my experiment as I depicted in last comment, when the temperature is low, ZERO still has the largest logit. Do you have some thoughts on this phenomenon?

---

> > > > > ### Author Response · Authors · 2018-11-02
> > > > > **Response to the follow-up**
> > > > >
> > > > > > "Could you please give an explanation?" (on the role of zero ops for edge selection)
> > > > > Since the zero op has been taken into account in the denominator of the edge strength (defined in sect. 2.4), edges with large weights on the zero ops are less likely to be selected.
> > > > >
> > > > > Our implementation follows the intuitions above. In particular, strengths of the zero ops are included for row-wise normalization of W (L154-155). The normalized W will then affect the output of L142 to determine the selected edges.
> > > > >
> > > > > > "Do you have some thoughts on this phenomenon?"
> > > > > It is tempting to replace our current discretization scheme with temperature annealing + argmax. However, we found it nontrivial to come up with a suitable annealing schedule to simultaneously ensure (1) the temperature is low enough to yield a near-discrete architecture (thus getting rid of the “mixing effect” that you are referring to) (2) the temperature is high enough so that \alpha does not get stuck at some suboptimal region, e.g., solution with lots of zeros. We leave more investigations on this direction as an interesting future work.

---

### Public Comment · (anonymous) · 2018-10-25
**A severe problem for the objective function, the loss is wrong**

Hello,

The key contribution of this work is to propose that architecture search can be carried on by gradient decent. That is great!

The solution of this work lies on relaxing "the categorical choice of a particular operation as a softmax over all possible operations". However, the objective (eq. 3) based on this relaxation is not equivalent to expectation of loss over all possible architectures. But the  expectation of loss over all possible architectures should be the correct metric to be optimized. Hence, I think the loss of DARTS does not make sense.

I do not have a hard feeling on this work. Instead, I appreciate the work. However, I just think the loss is not correct and want to discuss it here to make it clearer.

---

> ### Author Response · Authors · 2018-10-25
> **The criticism is invalid**
>
> Thank you for the comments. We respectfully disagree with your statement that “the loss is wrong.”. The reasons are as follows:
>
> (1) Our architecture encoding is deterministic and we don’t maintain any probability distribution over architectures. Hence “expectation of loss over all possible architectures” in your statement is not even well-defined, not to mention the statistical consistency.
> (2) eq. 3 is just the paraphrase of “finding a (deterministic) architecture that minimizes its final validation loss.”. No stochasticity is involved.
> (3) The continuous architecture \alpha is nothing but a high-dimensional hyperparameter. While bi-level optimization is new to the field of architecture search, formulations similar to eq. 3 have been well-studied for hyperparameter search [1,2,3].
> (4) Effectiveness of eq. 3 has been empirically verified by extensive experiments.
>
> [1] Dougal Maclaurin, David Duvenaud, and Ryan Adams. Gradient-based hyperparameter optimization through reversible learning. In ICML, pp. 2113–2122, 2015.
> [2] Fabian Pedregosa. Hyperparameter optimization with approximate gradient. In ICML, 2016.
> [3] Luca Franceschi, Paolo Frasconi, Saverio Salzo, and Massimilano Pontil. Bilevel programming for hyperparameter optimization and meta-learning. ICML, 2018.

---

### Public Comment · (anonymous) · 2018-11-02
**Regarding the claim: "able to discover both convolutional and recurrent networks"**

Hi!

Interesting work! This question is more on the particular writing style and not on the method.

 About the claim of being "able to discover both convolutional and recurrent networks" as mentioned in text, I don't think it's an accurate way to remark that. Given that here you search for a computation cell, and quoting from section 2.1 "The learned cell could either be stacked to form a convolutional network or recursively connected to form a recurrent network.".

In my opinion, this doesn't imply that the method discovered recurrence or convolutional architecture, but instead it was explicitly done by stacking cells in a recurrent manner or providing a convolution as candidate operation.   I would request the authors to reconsider their way of writing this and maybe say something like, "able to discover effective cells for use in convolutional and recurrent networks".

Thanks!

---

> ### Author Response · Authors · 2018-11-02
> **Good point!**
>
> Thanks for the suggestion. We will revise our writing accordingly.

---

### Public Comment · (anonymous) · 2018-11-08
**Details on evaluation**

Hello, authors!
To begin with, I am so impressed by this work because it is both simple and powerful.
However, I am curious about some details on your model evaluation.

According to this paper, there are 7 nodes within a cell for both search and evaluation, and 8 cells were used for search and 20 cells were used for evaluation. Here, could you specify derived model for evaluation in terms of the number of initial channels? Evaluated model in DARTS for state-of-the-art comparison for CIFAR10 does not seem to match with its reported number of parameters(2.9M or 3.3M) if the number of initial channels was kept the same with architecture search as 16. I think it should have fewer parameters than 2.9M or 3.3M if the number of initial channels was kept the same. Could you answer this?

Anyway, thanks for this amazing work!

---

### Public Comment · (anonymous) · 2018-11-22
**Doubt on FLOPS of DARTS**

Hi authors!

I enjoy your paper with awesome codes.

Here I have one question about FLOPS of DARTS on ImageNet in the mobile setting.
I have come to the conclusion that FLOPS of DARTS on ImageNet in the mobile setting is 585M/s, which conflicts with 574M/s provided in Table 3 of your paper.

Can the authors clarify this doubt? Thank you.

---

### Author Response · Authors · 2018-11-25
**Draft Update**

We thank all reviewers and public commenters for their feedback. The draft has been updated and major changes include:
+ Fixed some claims, typos and missing references.
+ Revised sect. 2.3 to better explain the motivation of our algorithm.
+ Revised sect. 2.4 to make the description of our discretization scheme more intuitive.
+ Highlighted the selection and evaluation costs on top of Table 1 & 2.
+ Added results of repeating ENAS for 8 times in Table 1.
+ Added results of simultaneously optimizing w and \alpha over the same set instead of two separate data splits in sect. 3.3.
+ Changes addressing the public comments.

---

### Public Comment · (anonymous) · 2018-12-16
**Another related work is missing**

One relevant NeurIPS paper this year, which shares the same high-level idea as this work and searches architectures in a continuous and differentiable space, is missing.

Neural Architecture Optimization
https://nips.cc/Conferences/2018/Schedule?showEvent=11750

---

### Public Comment · (anonymous) · 2018-12-18
**A severe problem on the searching for RNN architectures. The code provided by the authors is WRONG!**

Hi,

The work is quite interesting! After reading the paper carefully, I read the code provided by the authors on Github (https://github.com/quark0/darts). However, I found that the code for RNN searching is WRONG!

In ENAS (https://arxiv.org/abs/1802.03268), the paper mentioned: "In the example above, we note that for each pair of nodes j < ℓ, there is an independent parameter matrix W^(h)_{ℓ,j} . As shown in the example, by choosing the previous indices,
the controller also decides which parameter matrices are used. Therefore, in ENAS, all recurrent cells in a search space share the same set of parameters." That means if there are N nodes, there should be (1+2+...+N-1) weights, for each pair of nodes j<ℓ. This setting is reasonable, as if node ℓ is connected to different nodes, the connection matrices should be different.

HOWEVER, I find that DARTS use the SAME Weights if node ℓ is connected to different nodes!! In the code, there are only N connection weight Ws (https://github.com/quark0/darts/blob/master/rnn/model.py#L26), and the correct number of Ws should be (1+2+...+N-1).  This error is also confirmed by https://github.com/quark0/darts/blob/master/rnn/model_search.py#L28, where all the previous nodes j of node i share the same Ws[i], not Ws[i][j]. Using the SAME Weights is really wired and does not make sense! But the authors did not mention this point at all!

I think this is a bug in the code and the authors did not notice it. So the results is not convincing! I hope the authors should fix it the redo the experiments ASAP!

---

> ### Author Response · Authors · 2018-12-18
> **Our implementation is correct**
>
> This is not a bug, but a strategy to reduce the memory consumption when (1) parameters within all incoming ops are of the same shape and (2) we know that for each node only one of its predecessors will be retained (as in the case of RNNs) and the algorithm always has the option to zero out the others. It has been mentioned in sect. A.1.2, and we will explain it in more detail in the next revision.
>
> > "In the code, there are only N connection weight Ws"
> Like ENAS, each node in our derived recurrent cell has only a single predecessor, hence there should be N ops (W's) in total.

---

> > ### Public Comment · (anonymous) · 2018-12-19
> > **The strategy is really wired and not correct**
> >
> > 1. "This is not a bug, but a strategy to reduce the memory consumption":
> >       a）The size of Ws is only 300x300, it is not a big matrix that can cause OOM error on GPU. Why do you need to reduce the memory?
> >       b)   I have checked the implementation of ENAS (https://github.com/melodyguan/enas). For the RNN search, ENAS uses different W_{i,j} for different previous node j of node i. In ENAS, it even uses different W for different activations. That is to say, for node i, and there are 4 activations (Relu, Tanh, Sigmoid, Identity), there are (i-1)x4 connection weights for node i, as there are i-1 previous nodes and 4 activation functions. The implementation of ENAS makes sense. Since the weights should not be shared by different activations. So do the previous nodes. It is very very wired that different inputs use the same weights. If they are shared, there are no difference for the connection for different previous nodes, except the node itself.
> >      c) If the implementation of ENAS do not OOM (I tested ENAS code and it works), why do you use this wired strategy?
> >
> > 2. " It has been mentioned in sect. A.1.2"
> >      In this section, you mention: "The linear transformation parameters across all candidate operations on the SAME EDGE are shared". What does the "same edge" mean? I think the connection between node i and node j is an edge, and between node i and node k is an different edge (see figure 1). Edge (i-j) and Edge(i-k) are not the same edge, right? So in Sect. A. 1.2, I think the sentence means different ACTIVATION FUNCTIONS use the same connection matrix, and it does NOT mean any connection to node i uses the same weights. Hence, it is misleading.

---

### Author Response · Authors · 2018-12-19
**Generic Response**

Dear Reviewers,

In response to the negative anonymous comments that we have received, we would like to reiterate that our claims are valid, and the publicly available implementation is correct. Throughout the reviewing process, we have done our best to address all questions we have received, and we will strive to continue improving the paper.

---

> ### Public Comment · (anonymous) · 2018-12-20
> **Clearly, your implementation is not correct**
>
> At least, you should provide experimental results without the wired strategy. I think it is a big problem for the literature, it will make the future NAS work confuses on whether to use your "strategy".

---

### Meta-Review · Area_Chair1 · 2018-12-13
**Good paper. Accept.**

**Confidence:** 5
**Recommendation:** Accept (Poster)

**Metareview:**

This paper introduces a very simple but effective method for the neural architecture search problem. The key idea of the method is a particular continuous relaxation of the architecture representation to enable gradient descent-like differentiable optimization. Results are quite good. Source code is also available. A concern of the approach is the (possibly large) integrality gap between the continuous solution and the discretized architecture. The solution provided in the paper is a heuristic without guarantees.  Overall, this is a good paper. I recommend acceptance.